# Monogeneans from Catfishes in Lake Tanganyika. II: New Infection Site, New Record, and Additional Details on the Morphology of the Male Copulatory Organ of *Gyrodactylus transvaalensis* Prudhoe and Hussey, 1977

**DOI:** 10.3390/pathogens12020200

**Published:** 2023-01-28

**Authors:** Archimède Mushagalusa Mulega, Maarten Van Steenberge, Nikol Kmentová, Fidel Muterezi Bukinga, Imane Rahmouni, Pascal Mulungula Masilya, Abdelaziz Benhoussa, Antoine Pariselle, Maarten P. M. Vanhove

**Affiliations:** 1Laboratory Biodiversity, Ecology and Genome, Research Center Plant and Microbial Biotechnology, Biodiversity and Environment, Mohammed V University in Rabat, Rabat 10100, Morocco; 2Centre for Environmental Sciences, Research Group Zoology: Biodiversity & Toxicology, Hasselt University, 3590 Diepenbeek, Belgium; 3Département de Biologie, Centre de Recherche en Hydrobiologie, B.P. 73, Uvira, Democratic Republic of the Congo; 4OD Taxonomy and Phylogeny, Royal Belgian Institute for Natural Sciences, 1000 Brussels, Belgium; 5Unité d’Enseignement et de Recherche en Hydrobiologie Appliquée, Département de Biologie-Chimie, ISP/Bukavu, Bukavu, Democratic Republic of the Congo; 6ISEM, Université de Montpellier, CNRS, IRD, 34000 Montpellier, France

**Keywords:** Clariidae, *Clarias gariepinus*, gills, Gyrodactylidae, monogenea, parasite, Platyhelminthes, DRC, East Africa, failure to diverge

## Abstract

The ichthyofauna of Lake Tanganyika consists of 12 families of fish of which five belong to Siluriformes (catfishes). Studies on Siluriformes and their parasites in this lake are very fragmentary. The present study was carried out to help fill the knowledge gap on the monogeneans infesting the siluriform fishes of Lake Tanganyika in general and, more particularly, *Clarias gariepinus*. Samples of gills of *Clarias gariepinus* (Clariidae) were examined for ectoparasites. We identified the monogenean *Gyrodactylus transvaalensis* (Gyrodactylidae). This is the first time this parasite was found infecting gills. We are the first to observe a large spine in the male copulatory organ of this species and to provide measurements of its genital spines; this completes the description of the male copulatory organ, which is important in standard monogenean identification. This is the first monogenean species reported in *C. gariepinus* at Lake Tanganyika and the third known species on a representative of Siluriformes of this lake. It brings the total number of species of *Gyrodactylus* recorded in Lake Tanganyika to four. Knowing that other locations where this species has been reported are geographically remote from Lake Tanganyika, we propose a “failure to diverge” phenomenon for *G. transvaalensis*.

The distribution of the catfishes (Siluriformes) spans every continent except Antarctica. There are now 4094 recognized species distributed among 497 genera and 39 families in this order. In Africa’s freshwaters, Siluriformes are represented by 476 species in 51 genera and nine families [1]. This order contains some of the world’s most economically significant freshwater and brackish water fishes. In many nations, they make up a sizeable portion of inland fisheries. Several species are used in aquaculture and have been introduced around the globe. Additionally, many species are valuable to the aquarium industry, where they account for a substantial part of global trade [2]. A typical economically important species is *Clarias gariepinus* (Burchell, 1822) (Clariidae: Siluriformes), a strong, air-breathing catfish from Africa and the Middle East [3]. It is thought to be the freshwater fish species with the greatest natural geographic distribution in Africa. The most common fish species raised in sub-Saharan Africa seem to be African catfish [4], followed by ‘tilapia’ [5]. Despite the great diversity and economic importance of catfishes, there is a paucity of knowledge about their parasites [6,7,8].

Lake Tanganyika is the oldest [9,10] of the East African Rift Lakes. It has a unique ichthyofauna that also contains about 34 catfish species [11] belonging to five families: Bagridae, Clariidae, Claroteidae, Malapteruridae, and Mochokidae [12]. The lack of information on Tanganyikan and East African catfish parasites prompted us to study the monogeneans of representatives of Siluriformes in Lake Tanganyika [13].

Monogeneans (Platyhelminthes) are typically ectoparasitic on the gills, skin, or other external surfaces of fishes or on the exterior surfaces of other cold-blooded vertebrates. Additionally, there are several endoparasitic species. The adults can anchor themselves to the surfaces of their hosts using their conspicuous posterior attachment structure, or haptor. This noticeable organ is equipped with suckers, different types of hooks, or both, and often varies in form between species [14]. Monogenean species identification is not only based on the morphology of the sclerotized structures of the haptor but also on distal sclerotized sections of the reproductive systems: the male copulatory organ (MCO) and the vagina [15]. Monogeneans are known for their potential to negatively impact aquaculture. Recommended treatments include hydrogen peroxide (35%) [16] and formalin (25 ppm), and recommended chemicals for prevention are potassium permanganate (2 ppm), methylene blue (2 ppm), and sodium chloride (0.02%) [17]. This work is part of a series of studies we are conducting on the monogeneans of siluriform representatives in Lake Tanganyika. In particular, this study focuses on the fish *C. gariepinus*.

During a survey in Lake Tanganyika at the mouth of the Mutambala River (29°04.4042′ E, 04°16.4598′ S) in the Democratic Republic of the Congo (DRC), a single individual of *C. gariepinus* (Figure 1) was captured using a gill net. The specimen was killed by severing its spinal cord. Identification was carried out using the key of Fermon et al. [18].

To enable the study of its parasites, the gills of the specimen were stored in 96% ethanol. These gills were examined under a Wild Heerbrugg^®^ M8 binocular. We noted the presence of a single monogenean individual. The parasite specimen was recovered using an entomological needle and mounted on a slide in a drop of Hoyer’s medium [19]. The slide was covered with a coverslip to flatten the specimen. It was left for 24 h in a horizontal position before sealing the coverslip with glyceel [20]. Identification was performed using a Leica^®^ DM 2500 microscope equipped with a digital camera (Leica DMC 4500) and phase contrast. The slide was deposited at the Royal Museum for Central Africa, Tervuren, Belgium (accession number RMCA_VERMES_ 43681). Based on Přikrylová et al. [21] the specimen was identified as belonging to *Gyrodactylus transvaalensis* Prudhoe and Hussey, 1977 (Figure 2). It shares the sturdy hamulus with a root that is wide especially close to where it joins the shaft and finer towards its proximal end. As Přikrylová et al. [21] reported, the inner surface of the anchor root is flattened. These authors also demonstrated some intraspecific diversity, e.g., in the shape of the marginal hook sickle of this species, and also illustrated this for other congeners parasitizing catfishes, e.g., *Gyrodactylus rysavyi* Ergens, 1973 and *Gyrodactylus synodonti* Přikrylová, Blažek, and Vanhove, 2012. In the case of our specimen, the blunt, downward-pointing heel; the triangular toe; and the forward-pointing sickle proper that narrows at the point and of which the point bends at a nearly right angle correspond well with how Přikrylová et al. [21] characterize the marginal hook sickle shape of *G. transvaalensis*.

The measurements performed on the sclerified parts (Figure 3) of this monogenean follow [22] and are as follows (distances in µm): hamulus total length (HTL) = 44.9; hamulus root length (HRL) = 14.7; hamulus aperture distance (HAD) = 17.1; hamulus proximal shaft width (HPSW) = 6.9; hamulus point length (HPL) = 21.1; hamulus distal shaft width (HDSW) = 2.8; hamulus shaft length (HSL) = 27.9; hamulus inner curve length (HICL) = 1.5; the angle θ between the hamulus point tip through its base and the lower edge of the ventral bar articulation point on the hamulus ϴ = 44.0°; the angle between the hamulus point tip, its blade, and the vector describing its length and base α = 5.6°; the angle λ between the hamulus point tip through the vector describing its length to the inner curve of the hamulus and the lower edge of the ventral bar articulation point on the hamulus λ = 50.3°; marginal hook total length (MHTL) = 21.5; marginal hook shaft length (MHSL) = 18.2; marginal hook sickle length (MHSL) = 3.7; marginal hook sickle proximal width (MHSPW) = 4.4; marginal hook toe length (MHSTL) = 3.0; marginal hook sickle distal width (MHSDW) = 3.7; marginal hook aperture (MHAD) = 5.6; and marginal hook instep/arch height (MHIH) = 0.5. These measurements are within (or close to) the ranges reported by [21]; these authors also re-examined paratype specimens. Probably because of the fixation in Hoyer’s medium, the morphology of the haptoral dorsal and ventral bars was not discernible. The MCO has a diameter of 13.1 with at least six spines (visible) of average length 4.0 (3.6–5.0) arranged in a single row, and a single large one that is 6.2 long. While [21] mentioned a similar MCO diameter (14.5), they found the genital hard parts to be indiscernible on the studied specimen. Conversely, Prudhoe and Hussey [23] did not provide measurements regarding the MCO; they observed a single row with eight spines but saw no large spine. It is important to remember that the number of MCO spines in *Gyrodactylus* may vary within the same species [24,25]. We here provide the first report on the size of the MCO spines of *G. transvaalensis* and on the presence of a large spine in this species’ MCO.

This is the first time that this monogenean species is reported from this infection site, from the DRC, and from the Congo ichthyofaunal province. Indeed, *G. transvaalensis* was described in 1977 by Prudhoe and Hussey [23] on the skin of *C. gariepinus* captured at the confluence of the Elands and Olifants Rivers, about 17 km north of Marble Hall in the central Transvaal (24°48′45.94″ S; 29°21′28.94″ E), currently the Limpopo province, Republic of South Africa. The second report of this species originates from West Africa, where it was found on the fins of *Clarias anguillaris* (Linnaeus, 1758) at Mare Simenti, Niokolo Koba National Park (13°01.79′ N, 13°17.6′ W), Senegal [21] (Figure 4). Since the specimen of *G. transvaalensis* from Lake Tanganyika was found on the type host of this species, it is unlikely that we are dealing with an accidental infection. Additional sampling, however, is needed to ascertain whether this species commonly occurs on the region’s clariid catfishes or sporadically.

This study increases the number of species of monogenean flatworms known from representatives of Siluriformes in Lake Tanganyika from two [13] to three and the number of species belonging to *Gyrodactylus* in the lake from three [26] to four. In general, endemicity in Lake Tanganyika is high for a variety of taxa [27,28], including monogenean parasites [29,30]. Currently, 50 monogenean species are known from Lake Tanganyika (see Table 1), including two species of *Bagrobdella* Paperna, 1969; 40 of *Cichlidogyrus* Paperna, 1960; one of *Scutogyrus* Pariselle and Euzet, 1995; one of *Dolicirroplectanum* Kmentová, Gelnar and Vanhove, 2018; four of *Gyrodactylus* von Nordmann, 1832; and two of *Kapentagyrus* Kmentová, Gelnar and Vanhove, 2018.

Eighty six percent of monogenean species (43/50) are endemic to Lake Tanganyika. Non-endemic species include *Kapentagyrus limnotrissae* (Paperna, 1973), which was, together with its host, artificially introduced into Lake Kariba [44]. This species was first described by Paperna in 1973 in Lake Tanganyika, on *Limnothrissa miodon* (Boulenger, 1906), as *Ancyrocephalus limnotrissae*. Kmentová, Gelnar, and Vanhove subsequently redescribed it as *Kapentagyrus limnotrissae* (Paperna, 1973), which is specific to *L. miodon* [44]. *Gyrodactylus sturmbaueri* Vanhove, Snoeks, Volckaert, and Huyse, 2011 infects cichlids. It has been described from Lake Tanganyika on the gills (probably present on the fins and skin as well) of *Simochromis diagramma* (Günther, 1894) [26], but was subsequently reported on the gills of *Pseudocrenilabrus philander* (Weber, 1897) in South Africa and Zimbabwe [42]. *Dolicirroplectanum lacustre* (Thurston and Paperna, 1969) is a parasite of latid fishes throughout Africa [41]. *Cichlidogyrus mbirizei* Muterezi Bukinga, Vanhove, Van Steenberge, and Pariselle, 2012 has been described from the gills of *Oreochromis tanganicae* (Günther, 1894) from Lake Tanganyika [29] (on which the same authors also discovered dactylogyrid species already known from elsewhere [29]) but was later reported in Thailand [38], Malaysia [39], and China [40].

Eight species of monogeneans belonging to *Gyrodactylus* are known from African populations of *C. gariepinus* [46]. These are: *G. alberti* Paperna, 1973; *G. alekosi* Přikrylová, Blažek, and Vanhove, 2012; *G. clarii* Paperna, 1973; *G. gelnari* Přikrylová, Blažek, and Vanhove, 2012; *G. groschafti* Ergens, 1973; *G. rysavyi* Ergens, 1973; *G. transvaalensis* Prudhoe and Hussey, 1977; and *G. turkanaensis* Přikrylová, Blažek, and Vanhove, 2012.

**Figure 4 pathogens-12-00200-f004:**
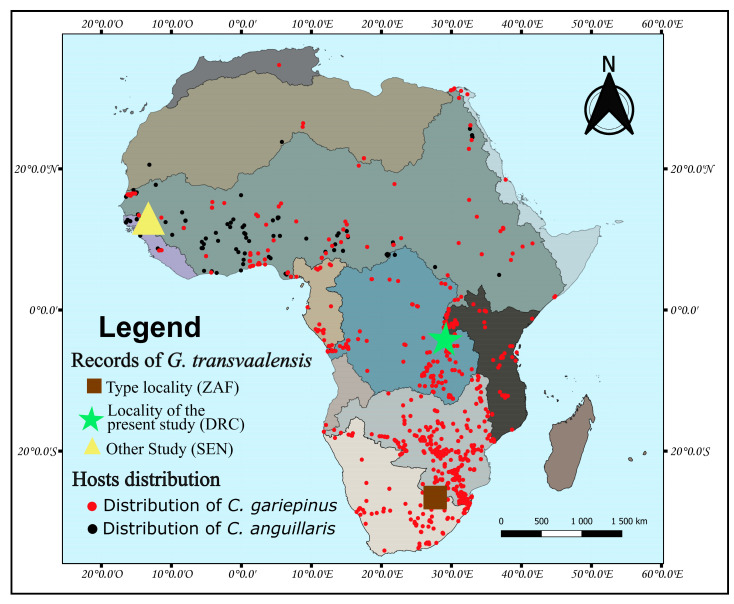
Records of *G. transvaalensis* and distribution range of its hosts. ZAF = South Africa, DRC = Democratic Republic of the Congo, and SEN = Senegal. The map was made with QGIS 3.16. Distribution data of *C. gariepinus* and *C. anguillaris* were downloaded from Fishbase (https://www.fishbase.se/map/OccurrenceMapList.php?genus=Clarias&species=gariepinus, 19 September 2022 and https://www.fishbase.se/map/OccurrenceMapList.php?genus=Clarias&species=anguillaris, 19 September 2022). Colors denote ichthyofaunal provinces, as in [47].

Molecular and morphological data indicate a large amount of variation in *C. gariepinus*, possibly hinting at unrecognized taxic diversity. Additionally, the species as currently defined is paraphyletic with respect to *C. anguillaris* and the species of *Bathyclarias* Jackson, 1959 [48,49,50]. Because of their often high species specificity, monogenean parasites can be valuable markers to solve problems in fish systematics [50]. Hence, we point out that three of the aforementioned parasite species, *G. gelnari*, *G. rysavyi*, and *G. transvaalensis*, are shared between *C. gariepinus* and *C. anguillaris* and that, apart from these three, no other species of *Gyrodactylus* are known in *C. anguillaris* [46]. Hence, a potential explanation of *G. transvaalensis* (as well as the other two species of *Gyrodactylus* mentioned above) being present on both *C. anguillaris* and *C. gariepinus* is that the fish host speciated and the parasite did not. This is labeled “failure to diverge” [51]. This was also suggested for *D. lacustre* [41], a monogenean infecting a widespread African freshwater fish and the species rendering it paraphyletic [52]. An alternative hypothesis that would require molecular data to be tested may be related to ongoing diversification in *G. transvaalensis*, driven by, for example, host preference or infection site preference. A further sampling of parasites of *C. gariepinus* and closely related species across their natural range should be undertaken. Combined with molecular studies on these parasites, this may, in view of their marker potential, help resolve uncertainties in the taxonomy and evolutionary history of these clariids.

## Figures and Tables

**Figure 1 pathogens-12-00200-f001:**
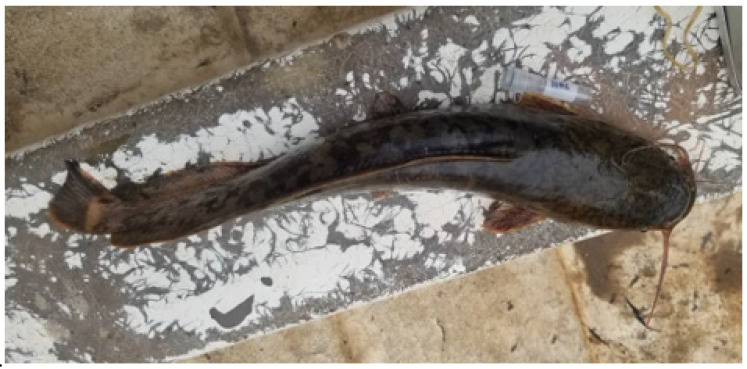
Examined specimen of *Clarias gariepinus* from Lake Tanganyika (Uvira, DRC); picture by Fidel Muterezi Bukinga.

**Figure 2 pathogens-12-00200-f002:**
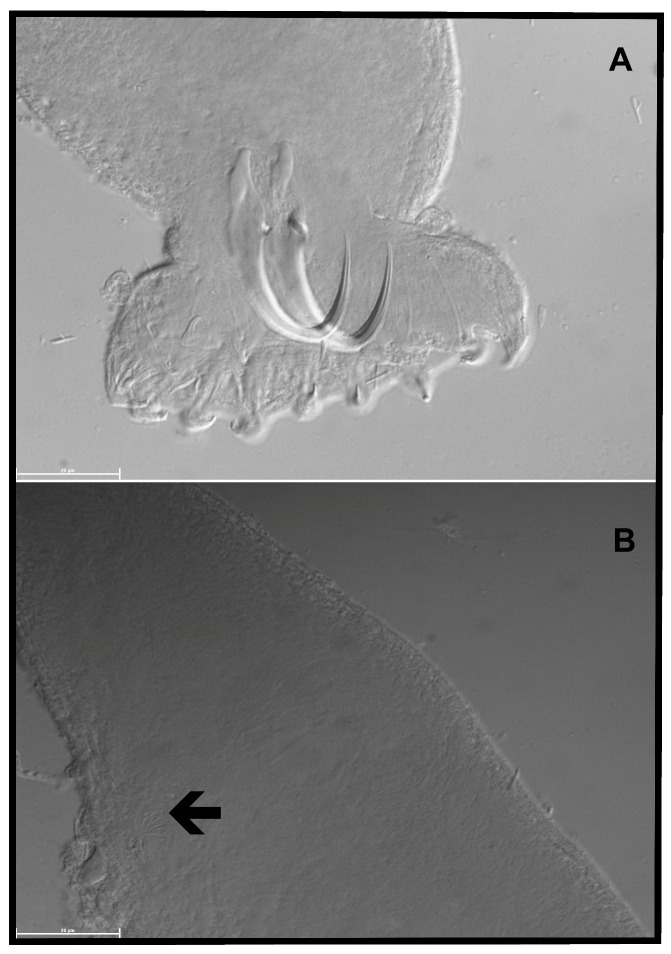
Photomicrographs of the haptoral (**A**) and genital (**B**) hard parts of *Gyrodactylus transvaalensis* from the gills of *Clarias gariepinus* captured in Lake Tanganyika off the mouth of the Mutambala River, DRC (RMCA_VERMES_ 43681). Scalebar measures 20 µm.

**Figure 3 pathogens-12-00200-f003:**
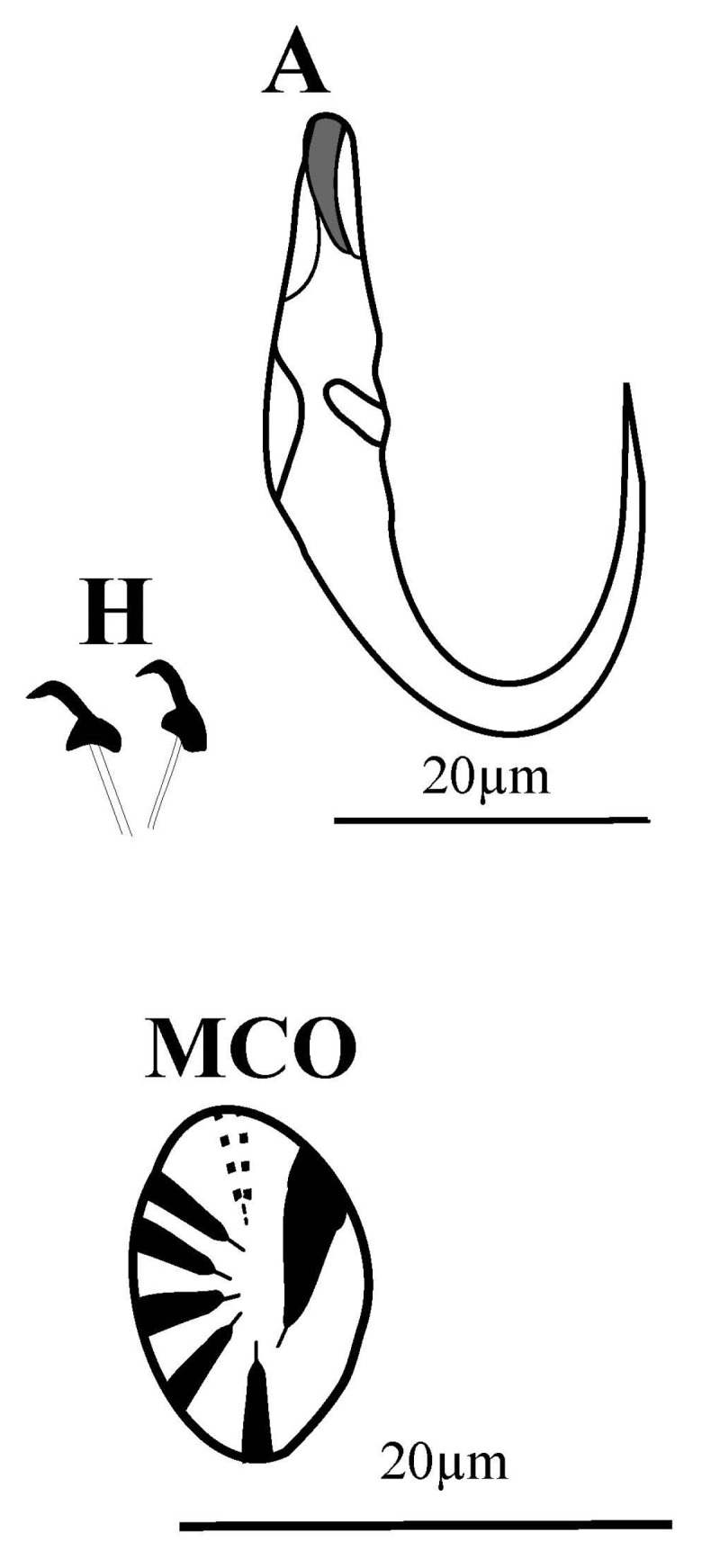
Line drawings of haptoral hard parts of *Gyrodactylus transvaalensis* (RMCA_VERMES_ 43681) (A = hamulus, H = marginal hooks, MCO = male copulatory organ).

**Table 1 pathogens-12-00200-t001:** Alphabetical list of monogenean species reported in Lake Tanganyika. The “Reference” column indicates the publication in which the species is reported from Lake Tanganyika, except for those marked by *, which indicates the publications that reported the same species outside of Lake Tanganyika. Species not endemic to Lake Tanganyika are listed in bold.

Species	Authors of Species	Reference
*Bagrobdella vanhovei*	Mushagalusa Mulega and Pariselle, 2022	[13]
*Bagrobdella vansteenbergei*	Mushagalusa Mulega and Pariselle, 2022	[13]
*Cichlidogyrus adkoningsi*	Rahmouni, Vanhove, and Šimková, 2018	[31]
*Cichlidogyrus antoineparisellei*	Rahmouni, Vanhove, and Šimková, 2018	[31]
*Cichlidogyrus aspiralis*	Rahmouni, Vanhove, and Šimková, 2017	[32]
*Cichlidogyrus attenboroughi*	Kmentová et al., 2016	[33]
*Cichlidogyrus banyankimbonai*	Pariselle and Vanhove, 2015	[34]
*Cichlidogyrus brunnensis*	Kmentová et al., 2016	[33]
*Cichlidogyrus buescheri*	Pariselle and Vanhove, 2015	[30]
*Cichlidogyrus casuarinus*	Pariselle, Muterezi Bukinga and Vanhove, 2015	[35]
*Cichlidogyrus centesimus*	Vanhove, Volckaert, and Pariselle, 2011	[36]
*Cichlidogyrus discophonum*	Rahmouni, Vanhove, and Šimková, 2017	[32]
*Cichlidogyrus evikae*	Rahmouni, Vanhove, and Šimková, 2017	[32]
*Cichlidogyrus frankwillemsi*	Pariselle and Vanhove, 2015	[34]
*Cichlidogyrus franswittei*	Pariselle and Vanhove, 2015	[34]
*Cichlidogyrus georgesmertensi*	Pariselle and Vanhove, 2015	[34]
*Cichlidogyrus gillardinae*	Muterezi Bukinga et al., 2012	[29]
*Cichlidogyrus gistelincki*	Gillardin et al., 2012	[37]
*Cichlidogyrus glacicremoratus*	Rahmouni, Vanhove, and Šimková, 2017	[32]
*Cichlidogyrus habluetzeli*	Rahmouni, Vanhove, and Šimková, 2018	[31]
** *Cichlidogyrus halli* ** *Cichlidogyrus irenae*	(Price and Kirk, 1967) Gillardin et al., 2012	[29] [37]
*Cichlidogyrus jeanloujustinei*	Rahmouni, Vanhove, and Šimková, 2017	[32]
*Cichlidogyrus koblmuelleri*	Rahmouni, Vanhove, and Šimková, 2018	[31]
*Cichlidogyrus makasai*	Vanhove, Volckaert, and Pariselle, 2011	[36]
*Cichlidogyrus masilyai*	Rahmouni, Vanhove, and Šimková, 2018	[31]
** *Cichlidogyrus mbirizei* **	Muterezi Bukinga et al., 2012	[29]
		* [38]
		* [39]
		* [40]
*Cichlidogyrus milangelnari*	Rahmouni, Vanhove, and Šimková, 2017	[32]
*Cichlidogyrus mulimbwai*	Muterezi Bukinga et al., 2012	[29]
*Cichlidogyrus muterezii*	Pariselle and Vanhove, 2015	[34]
*Cichlidogyrus muzumanii*	Muterezi Bukinga et al., 2012	[29]
*Cichlidogyrus nshomboi*	Muterezi Bukinga et al., 2012	[29]
*Cichlidogyrus pseudoaspiralis*	Rahmouni, Vanhove, and Šimková, 2017	[32]
*Cichlidogyrus raeymaekersi*	Pariselle and Vanhove, 2015	[34]
*Cichlidogyrus rectangulus*	Rahmouni, Vanhove, and Šimková, 2017	[32]
*Cichlidogyrus salzburgeri*	Rahmouni, Vanhove, and Šimková, 2018	[31]
*Cichlidogyrus schreyenbrichardorum*	Pariselle and Vanhove, 2015	[30]
*Cichlidogyrus sergemorandi*	Rahmouni, Vanhove, and Šimková, 2018	[31]
*Cichlidogyrus steenbergei*	Gillardin et al., 2012	[37]
*Cichlidogyrus sturmbaueri*	Vanhove, Volckaert, and Pariselle, 2011	[36]
*Cichlidogyrus vandekerkhovei*	Vanhove, Volckaert, and Pariselle, 2011	[36]
*Cichlidogyrus vealli*	Pariselle and Vanhove, 2015	[30]
** *Dolicirroplectanum lacustre* **	(Thurston and Paperna, 1969)	[41]
** *Gyrodactylus sturmbaueri* **	Vanhove, Snoeks, Volckaert, and Huyse, 2011	[26]
		* [42]
*Gyrodactylus thysi*	Vanhove, Snoeks, Volckaert, and Huyse, 2011	[26]
** *Gyrodactylus transvaalensis* **	Prudhoe and Hussey, 1977	* [23]
		* [21]
		Present study
*Gyrodactylus zimbae*	Vanhove, Snoeks, Volckaert, and Huyse, 2011	[26]
** *Kapentagyrus limnotrissae* **	(Paperna, 1973)	[43]
		[44]
		* [45]
*Kapentagyrus tanganicanus* ** *Scutogyrus gravivaginus* **	Kmentová et al., 2018 (Paperna and Thurston, 1969)	[44] [29]

## Data Availability

The voucher specimen was deposited in the Royal Museum for Central Africa (Tervuren, Belgium) under accession number RMCA_VERMES_ 43681. The observations here reported were posted on iNaturalist under https://www.inaturalist.org/observations/139872031 (host) and https://www.inaturalist.org/observations/139872032 (parasite).

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
