# Peer review of "Monogeneans from Catfishes in Lake Tanganyika. II: New Infection Site, New Record, and Additional Details on the Morphology of the Male Copulatory Organ of Gyrodactylus transvaalensis Prudhoe and Hussey, 1977"

_pathogens, 2023, doi:10.3390/pathogens12020200_

Round 1

Reviewer 1 Report

This paper aims to contribute novel epidemiological information (infection site on host, geographic locality and morphological features) regarding the parasite Gyrodactylus transvaalensis from Clarias gariepinus.  The novelty of these observations and the broader hypothesis regarding the divergence of this parasite largely rely on the accurate identification of the parasite in question.  In this case the entire manuscript is based on a single specimen from a single fish host.  Micrographs (Figure 2) clearly illustrate that this individual is not ideally flattened or positioned on the slide to optimally observe or measure key diagnostic taxonomic features required to identify these parasites.  Despite this, the sclerite morphometrics were reported as being within range for G. transvaalensis as per Prikrylova et al 2012.  Although this may be accurate, this would appear to also be true for G. alekosi and G. nigritae of Prikrylova et al 2012 as well.  Despite being able to identify this specimen as belonging to the Genus Gyrodactylus I am of the opinion that the evidence presented here is not sufficient to conclude on the identity of the species.  For instance, in Gyrodactylus taxonomy, the marginal hooklet sickle morphology is considered as being quite conserved in species.  The similarity in hooklet sickle morphology is clearly illustrated for the two known populations of G. transvaalensis in Fig 2G & H of Prikrylova et al 2012 and is described as being "ovate in profile with a slightly triangular toe and blunt heel pointing downwards.  The sickle proper rises at slightly forward angle; sickle point narrow and angled almost perpendicularly, just extending beyond toe." This is vastly different from what is illustrated in Fig 3H.  In fact, the marginal hooklet sickle illustrated here does not closely resemble that of any of the species presented in Fig 2 of Prikrylova et al 2012.  Further to this, the specimen does not allow for the observation of the Ventral and Dorsal bars but features of the MCO not previously described for G transvaalensis are presented here.   

Author Response

This paper aims to contribute novel epidemiological information (infection site on host, geographic locality and morphological features) regarding the parasite Gyrodactylus transvaalensis from Clarias gariepinus.  The novelty of these observations and the broader hypothesis regarding the divergence of this parasite largely rely on the accurate identification of the parasite in question.  In this case the entire manuscript is based on a single specimen from a single fish host.  Micrographs (Figure 2) clearly illustrate that this individual is not ideally flattened or positioned on the slide to optimally observe or measure key diagnostic taxonomic features required to identify these parasites.  Despite this, the sclerite morphometrics were reported as being within range for G. transvaalensis as per Prikrylova et al 2012.  Although this may be accurate, this would appear to also be true for G. alekosi and G. nigritae of Prikrylova et al 2012 as well. 

Response: The identification of this species was made based on the diagnosis made by Prudhoe and Hussey, and later by PÅ™ikrylová and colleagues. Based on the latter authors, the two species the reviewer mentions (G. alekosi and G. nigritae) cannot be linked to the specimen presented in this study. Indeed, the total size of the anchor (HTL) and the length of its point (HPL) and root (HRL), exclude the possibility of identifying our specimen either as G. alekosi (HTL 50.5-53; HPL 25-27.5; HRL 25-29) or G. nigritae (HTL 43-51.5; HPL 25-28.3; HRL 19-30) (all measurements in µm). As the reviewer acknowledges, for our specimen, measurements correspond to those of the type of G. transvaalensis reexamined by PÅ™ikrylová et al. 2012 (HTL 41.5-45; HPL 20.5-23; HRL 14.5-17) and the newly discovered population of G. transvaalensis of PÅ™ikrylová et al (HTL 41.5-44.5; HPL 20-23; HRL 20-22).  However, these numbers also make clear there is no overlap in HTL or HRL between G. transvaalensis and G. alekosi, and no overlap in HPL between G. transvaalensis and either of the two species mentioned by the reviewer. Moreover, in comparison to G. transvaalensis the marginal hook sickles of G. alekosi and G. nigritae have a blunter toe (as opposed to the triangular toe in G. transvaalensis) and a more regularly curved point (as opposed to the right angle observed in the sickle point of the marginal hook of G. transvaalensis).

Despite being able to identify this specimen as belonging to the Genus Gyrodactylus I am of the opinion that the evidence presented here is not sufficient to conclude on the identity of the species.  For instance, in Gyrodactylus taxonomy, the marginal hooklet sickle morphology is considered as being quite conserved in species.  The similarity in hooklet sickle morphology is clearly illustrated for the two known populations of G. transvaalensis in Fig 2G & H of Prikrylova et al 2012 and is described as being "ovate in profile with a slightly triangular toe and blunt heel pointing downwards.  The sickle proper rises at slightly forward angle; sickle point narrow and angled almost perpendicularly, just extending beyond toe." This is vastly different from what is illustrated in Fig 3H.  In fact, the marginal hooklet sickle illustrated here does not closely resemble that of any of the species presented in Fig 2 of Prikrylova et al 2012. 

Response: We apologise for the confusion caused, and have now improved the quality of the illustrations, and provided a qualitative comparison of the marginal hook sickle shape between our specimen and the specimens reported in Prikrylova et al. (2012).

Further to this, the specimen does not allow for the observation of the Ventral and Dorsal bars but features of the MCO not previously described for G transvaalensis are presented here.

Response: We have explained in the manuscript why the bars were not observed, and present these additional features of the MCO as one of the main findings of our study.

Reviewer 2 Report

Line 22: parasitizing is an odd word. I prefer monogeneans infestation.

Line 24: revise as specimens of gills

Please write the fish name in abbreviation C. gariepinus. Full name should be used once at each particular section

Line 25: It is a fact some monogeneans are strictly gill parasite, but the majority could affect fish skin and external surfaces, please consider

Line 29: reported in not on

Line 34: select only five keywords

Although the authors provide comprehensive and valuable facts about monogeneasis, they still need to go deeply with their prevalence therapeutic and prophylactic treatments. I recommend writing one paragraph about that using the following paper as a reference

Trichodinids and monogeneans infestation among Nile tilapia hatcheries in Egypt: prevalence, therapeutic and prophylactic treatments https://doi.org/10.1007/s10499-020-00537-w

Line 61: please revise as follow: is not only based on the morpholo……..

Line 67: the study was only conducted on a single individual of C. gariepinus. Is it representative to ensure the endemic infestation in that lake, or is it like a sporadic case, please explain

In figure 1: It is preferable to present the gills lesion along with the whole body of the examined fish, as the severity of infection and marbling gill appearance if fully dependent on the number of parasites and their intensity.

What about adding one figure using SEM microscope. It will be of great impact for all readers

Author Response

Line 22: parasitizing is an odd word. I prefer monogeneans infestation.

Response: We agree

Line 24: revise as specimens of gills

Response: we followed this suggestion but opted for the word “sample” instead of “specimens”, to avoid confusion with “specimen” as in “individual animal”.

Please write the fish name in abbreviation C. gariepinus. Full name should be used once at each particular section

Response: We agree and have only written the name in full at the beginning of a new section or in figure legends.

Line 25: It is a fact some monogeneans are strictly gill parasite, but the majority could affect fish skin and external surfaces, please consider

Response: Many thanks for this remark. The current manuscript focuses on gill parasites; specifically, it reports the presence of a parasite species that was never before reported from gills. Therefore, we preferred not to stress non-gill parasites in the abstract. However, we agree with your remark and have included a statement to this effect on lines 56-57 of the revised manuscript.

Line 29: reported in not on

We agree

Line 34: select only five keywords

Response: Since the journal instructions offer us the possibility to list up to 10 keywords, for the visibility of our study, we have chosen to use as many keywords as possible.

Although the authors provide comprehensive and valuable facts about monogeneasis, they still need to go deeply with their prevalence therapeutic and prophylactic treatments. I recommend writing one paragraph about that using the following paper as a reference

Trichodinids and monogeneans infestation among Nile tilapia hatcheries in Egypt: prevalence, therapeutic and prophylactic treatments https://doi.org/10.1007/s10499-020-00537-w

Response: Many thanks for this suggestion; we have included this information on lines 64-67.

Line 61: please revise as follow: is not only based on the morpholo……..

Response: We agree.

Line 67: the study was only conducted on a single individual of C. gariepinus. Is it representative to ensure the endemic infestation in that lake, or is it like a sporadic case, please explain

Response:  This study does not have an ecological or epidemiological purpose. The main objective is to report the presence of this species in the Lake Tanganyika basin, in this host species, and to provide additional information on its site of infection and the morphology of its male copulatory organ. We agree with the reviewer that this indeed may be a sporadic case, and have included a statement about this on lines 143-146.

In figure 1: It is preferable to present the gills lesion along with the whole body of the examined fish, as the severity of infection and marbling gill appearance if fully dependent on the number of parasites and their intensity.

Response:   Figure 1 merely serves as a proof of host identification. Gill histopathology is out of the scope of this study; also, lesions are not expected in such low-intensity infestation.

What about adding one figure using SEM microscope. It will be of great impact for all readers

Response: With only one individual already fixed, it is unfortunately no longer possible to take photos with the SEM.

Reviewer 3 Report

Overall the paper is well written and the study design is fine for the most part. However, only one worm was found, which significantly impairs the ability for it to be identified. For example, molecular data was not obtained, the ventral bar was not observed, and morphologically the specimen is not exactly the same as reported in previous accounts. Just because the measurements overlap does not mean it is indeed the same species. This parasite also infects a different site on the fish (gills herein; fins in PÅ™ikrylová; and skin in the original description). Furthermore, the photos of the anchors and drawings of marginal hooks presented for G. transvaalensis in PÅ™ikrylová appear different than those presented herein. 

In order to fix the paper, a more thorough comparison between the specimen and the paratype for G. transvaalensis is necessary. Since there are not any molecular data for the parasite, the morphological data must be more convincing. Measurements are not enough in this case. In my opinion, side by side photographic comparisons (not drawings) of the specimens collected and the paratype are necessary to show that they are indeed the same parasite (I know this may be difficult with an old paratype but it would be the most conclusive way to show this). 

A few minor comments related to lines within the manuscript:

Line 81: Expand on how the parasite was identified "based on PÅ™ikrylová et al. "

Figure 2. The anchors do not appear as thick as those in PÅ™ikrylová et al. 

Figure 3: The marginal hook also looks different from those presented in PÅ™ikrylová

Line 167: Maybe these parasites have slightly diverged and that's the reason they are infecting different tissues than those presented in the original description and PÅ™ikrylová. They could be cryptic species and genetic data would be needed to prove it. 

In short, I am not convinced that the parasites presented in the original description, PÅ™ikrylová et al. and this paper are the same species. High quality photographs of taxonomically important features as well as molecular data from these sites would definitely confirm this possibility. 

Author Response

Overall the paper is well written and the study design is fine for the most part.

Response: we thank the reviewer for this positive appraisal of our work. 

However, only one worm was found, which significantly impairs the ability for it to be identified. For example, molecular data was not obtained, the ventral bar was not observed, and morphologically the specimen is not exactly the same as reported in previous accounts.

Response: as the reviewer states below themselves, molecular data cannot come to our aid here since the species has never been sequenced by previous authors. The fixative we used may indeed render the ventral bar hard to observe (as we mentioned on lines 120-122), while being ideally suited for observing anchors and marginal hooks in ethanol-preserved specimens. The fact that the specimen is not exactly morphologically the same seems a consequence of the intraspecific diversity commonly observed in members of Gyrodactylus, and also reported for this species, by PÅ™ikrylová et al. (2012, Parasitology Research). We have incorporated this important aspect now in the manuscript on lines 90-93 and thank the reviewer for pointing out that we did not mention the challenge of intraspecific diversity.

Just because the measurements overlap does not mean it is indeed the same species.

Response: we entirely agree with the reviewer and we regret having given the impression that we would find the measurements sufficient to identify a species of Gyrodactylus. We have now added a qualitative characterization (in addition to the quantitative, measurement-based) of the anchor and marginal hook in the light of how PÅ™ikrylová et al. (2012) reported their shapes on lines 88-97.

This parasite also infects a different site on the fish (gills herein; fins in PÅ™ikrylová; and skin in the original description).

Response: We consider this one of our main findings.

Furthermore, the photos of the anchors and drawings of marginal hooks presented for G. transvaalensis in PÅ™ikrylová appear different than those presented herein. 

Response: We have improved the images presented in our manuscript, and remain convinced of our identification for the reasons explained in our manuscript.

In order to fix the paper, a more thorough comparison between the specimen and the paratype for G. transvaalensis is necessary. Since there are not any molecular data for the parasite, the morphological data must be more convincing. Measurements are not enough in this case. In my opinion, side by side photographic comparisons (not drawings) of the specimens collected

and the paratype are necessary to show that they are indeed the same parasite (I know this may be difficult with an old paratype but it would be the most conclusive way to show this). 

Response: Sequencing would be of no use here, as the species has never been sequenced. We improved the photos and drawings using a phase contrast microscope. Unfortunately, we cannot access the paratype within the ten days timeframe we have been given to submit the revised version of the article. Luckily, PÅ™ikrylová et al. (2012) provided high-quality images of the haptoral hard parts of Gyrodactylus transvaalensis so that we can identify our specimen with certainty on that basis.

A few minor comments related to lines within the manuscript:

Line 81: Expand on how the parasite was identified "based on PÅ™ikrylová et al. "

Response:  Sorry for this omission. We have now expanded on this on lines 88-97.

Figure 2. The anchors do not appear as thick as those in PÅ™ikrylová et al. 

Figure 3: The marginal hook also looks different from those presented in PÅ™ikrylová

Response: We improved the photos using the phase contrast microscope, and have now elaborated how our specimen corresponds to how PÅ™ikrylová et al. diagnose this species.

Line 167: Maybe these parasites have slightly diverged and that's the reason they are infecting different tissues than those presented in the original description and PÅ™ikrylová. They could be cryptic species and genetic data would be needed to prove it. 

Response: It is impossible to verify this hypothesis since the species has never been sequenced; we have mentioned this hypothesis now on lines 201-203.

In short, I am not convinced that the parasites presented in the original description, PÅ™ikrylová et al. and this paper are the same species. High quality photographs of taxonomically important features as well as molecular data from these sites would definitely confirm this possibility. 

Round 2

Reviewer 3 Report

The manuscript has been improved, but one point that should be clarified is why there are exaggeratedly long anchor roots in Prudhoe and Hussey 1976, but they are much shorter in PÅ™ikrylová and in the present study. PÅ™ikrylová does examine the root of the holotype and says it’s the same as the specimen they found but doesn’t explain why the measurements and drawings in Prudhoe and Hussey report a much larger anchor (~75 um). What did they see? Maybe this cannot be addressed but it is a glaring discrepancy between the original description and the data presented in PÅ™ikrylová and the manuscript which is under review.

I have a few minor suggestions.

Line 109: “for their potential to negatively impact aquaculture.”

Line 112: Since you’re listing treatments for Gyrodactylus, it would be good to include Hydrogen Peroxide.

Line 246: Delete “rather”

Line 303: Add “related to” before “ongoing”

Figure 2: Change color (black or white – red just doesn’t seem right) and font (the current font doesn’t look the greatest). Also please at least make the images the same size or crop them and add borders (which would look nicer).

Figure 4: I forgot to mention this in the previous review but this is a really nice Figure!

Author Response

 We took into consideration all the comments and observations of reviewer number 3 (the only one left) as you can see below.

Reviewer: “I have a few minor suggestions.”

Line 109: “for their potential to negatively impact aquaculture.”

Response: We agree

Reviewer: Line 112: Since you’re listing treatments for Gyrodactylus, it would be good to include Hydrogen Peroxide.

Response: We agree

Reviewer: Line 246: Delete “rather”

Response: We agree

Reviewer: Line 303: Add “related to” before “ongoing”

Response: We agree

Reviewer: Figure 2: Change color (black or white – red just doesn’t seem right) and font (the current font doesn’t look the greatest). Also please at least make the images the same size or crop them and add borders (which would look nicer).

Response: We agree

Reviewer: Figure 4: I forgot to mention this in the previous review but this is a really nice Figure!

Thank you for your compliments.